# Genome-Wide Analysis of lncRNA and mRNA Expression in the Uterus of Laying Hens during Aging

**DOI:** 10.3390/genes14030639

**Published:** 2023-03-03

**Authors:** Guang Li, Xinyue Yang, Junyou Li, Bingkun Zhang

**Affiliations:** 1State Key Laboratory of Animal Nutrition, China Agricultural University, Beijing 100193, China; 2Graduate School of Agricultural and Life Sciences, The University of Tokyo, Tokyo 319-0206, Japan

**Keywords:** uterus, lncRNA, mRNA, co-expression network, aging

## Abstract

Eggshell plays an essential role in preventing physical damage and microbial invasions. Therefore, the analysis of genetic regulatory mechanisms of eggshell quality deterioration during aging in laying hens is important for the biosecurity and economic performance of poultry egg production worldwide. This study aimed to compare the differences in the expression profiles of long non-coding RNAs (lncRNAs) and mRNAs between old and young laying hens by the method of high-throughput RNA sequencing to identify candidate genes associated with aging in the uterus of laying hens. Overall, we detected 176 and 383 differentially expressed (DE) lncRNAs and mRNAs, respectively. Moreover, functional annotation analysis based on the Gene Ontology (GO) and Kyoto encyclopedia of genes and genomes (KEGG) databases revealed that DE-lncRNAs and DE-mRNAs were significantly enriched in “phosphate-containing compound metabolic process”, “mitochondrial proton-transporting ATP synthase complex”, “inorganic anion transport”, and other terms related to eggshell calcification and cuticularization. Through integrated analysis, we found that some important genes such as *FGF14*, *COL25A1*, *GPX8*, and *GRXCR1* and their corresponding lncRNAs were expressed differentially between two groups, and the results of quantitative real-time polymerase chain reaction (qPCR) among these genes were also in excellent agreement with the sequencing data. In addition, our study found that *TCONS_00181492, TCONS_03234147,* and *TCONS_03123639* in the uterus of laying hens caused deterioration of eggshell quality in the late laying period by up-regulating their corresponding target genes *FGF14*, *COL25A1*, and *GRXCR1* as well as down-regulating the target gene *GPX8* by *TCONS_01464392*. Our findings will provide a valuable reference for the development of breeding programs aimed at breeding excellent poultry with high eggshell quality or regulating dietary nutrient levels to improve eggshell quality.

## 1. Introduction

As one of the most affordable sources of available animal protein, eggs are widely favored by consumers around the world, and indeed, eggs dominate commercial markets in many countries [1]. The quality of eggshells, which is both of biological interest and economic importance to the poultry industry, has always been a major concern for the quality and safety of egg products. However, the huge economic loss caused by a deterioration in eggshell quality has been a pressing problem for the egg industry, which has become more acute in the late laying period. The incidence of cracked and broken eggs has been reported to be up to 12–20% in the late laying period, which is one of the key obstacles to extending the laying cycle of laying hens [2]. Most notably, it has been observed that the incidence of damaged and thin-shelled eggs is increased, and the egg production rate is reduced, with the aging of laying hens [3,4]. Therefore, understanding the transcriptomic regulation of eggshell quality with respect to aging is of great economic and biological importance. Further, the deterioration of eggshell quality is directly related to an increased risk of foodborne disease for consumers [5]. Therefore, improving eggshell quality is critically important for the poultry industry and human health.

As is well known, the uterus (shell gland) is composed of glandular and luminal epithelial cells whose secretions promote the biomineralization of the eggshell. The process of eggshell formation is separated into three essential stages: the initiation of crystal growth, linear crystal growth, and the termination of mineralization [6]. An eggshell is a highly ordered structure consisting of a bi-layered membrane (inner and outer), mammillary layer, palisade layer, vertical crystal layer, and cuticle layer. However, eggshell in the late laying period has a lower breaking strength as well as greater variation in structural properties such as thickness, grain morphology, and crystal texture [7]. In addition, the incidence of confluence and early fusion in the mammillary layer decreased with age [8]. Other research has also found that the decreased gland density and a shift in the balance between estrogen receptor α and estrogen receptor β in the shell gland could be critical factors explaining the age-related changes in eggshells [9]. Feng et al. (2020) utilized uterine transcriptome analysis to identify that altered gene expression of matrix proteins in the uterus of laying hens in the late phase of production may trigger age-related impairment of eggshell ultrastructure and its mechanical properties [10]. Therefore, we speculate that deteriorations in eggshell ultrastructure and quality in the late phase of production may be related to age-related abnormalities of gene expression in the uterus. It was supported by the findings that the expression of genes encoding ion transporters and matrix proteins (ATP2A2, SCNN1G, CA2, and ovocalyxin-36) varied with the age of laying hens [11].

Long non-coding RNAs (LncRNAs), with sizes >200 nt, are not translated into proteins [12], are found in both the nucleus and cytoplasm and have received much attention over the past several years. LncRNAs are involved in various aspects of disease and cell and molecular biology, such as cancer, the immune response, neurological and cardiovascular system disorders, cell cycle regulation, cell differentiation, X chromosome inactivation, genomic imprinting, transcriptional control, and epigenetic regulation [13,14,15]. LncRNA, which can regulate target genes in *cis* and *trans*, are key regulatory molecules. *Cis*-acting lncRNAs regulate the expression of target genes that are located at neighboring genomic loci, whereas *trans*-acting lncRNAs can regulate the expression of transcripts that are located at distal chromosomal loci [16]. The important roles of lncRNAs in the development of different organs and tissue types have been highlighted by many studies. For example, the studies reported here reveal a potential role for the lncRNA *MHM* and *MHM* in regulating embryonic growth and gonadal development [17,18]. The loss of lncRNA *MHM* expression in hens can cause asymmetric development of the ovary, and the loss of lncRNA *MHM* expression in males may result in decreased expression of the *DMRT1* gene in testis [17]. Similarly, the lncRNA *alphaGT* controls the expression of the *α-globin* gene from the embryo to the adult and plays a key role in chicken development [19]. However, there is almost no research on the combination of lncRNAs and shell gland development.

To date, however, the potential molecular mechanisms regulating eggshell quality by lncRNAs in the shell gland during aging in laying hens have not been clearly defined. Therefore, in this study, we performed a transcriptomic analysis of the shell gland among old and young laying hens to identify potential mRNAs and lncRNAs that regulate the aging of the shell gland in laying hens. We captured both lncRNAs and mRNAs from fragmented or intact RNA samples to compare whole transcriptomes of old and young chicken shell glands at unprecedented depth. Then, the differentially expressed (DE)-lncRNAs were used in bioinformatics analyses to predict *cis*- and *trans*-target genes and to construct lncRNA-mRNA co-expression interaction networks. Next, Gene Ontology (GO) and Kyoto Encyclopedia of Genes and Genomes (KEGG) functional analyses were performed to investigate the related roles of differentially expressed genes (DEGs). Our results may help to reveal key mRNAs and lncRNAs involved in the age-related deterioration of eggshell quality and provide potential candidate biomarkers for improving and intervening in eggshell quality in the late laying stage of laying hens.

## 2. Material and Methods

### 2.1. Ethics Statement

All the work using animals was approved by the Animal Care and Use Committee of China Agricultural University (No. AW42110202-2-2). All procedures were conducted in accordance with the institutional animal ethics guidelines set by the Ministry of Agriculture and Rural Affairs of the People’s Republic of China.

### 2.2. Animal and Sample Collection

The eight Hy-Line Brown commercial laying hens used in this study were purchased from Zhuozhou Chicken Farm. These hens were randomly assigned to old (60-week-old, *n* = 4) and young (31-week-old, *n* = 4) groups. All birds included in this study were raised on the same diet and managed conditions until slaughtered. Eighteen hours after laying the egg, animals were euthanized with carbon dioxide (approximately 5 min in a small container gassed with carbon dioxide from a compressed gas cylinder). Then, we collected the eggshell glands of each hen from the same pre-determined site and immediately flash frozen in liquid nitrogen and then stored them at −80 °C until RNA extraction.

### 2.3. Total RNA Extraction

Total RNA was extracted from shell gland tissues using TRIzol reagent according to the manufacturer’s instructions (Invitrogen Life Technologies, Carlsbad, CA, USA). RNA integrity was ascertained by 1.5% agarose gel electrophoresis, and the purity and concentration of the RNA were measured by spectrophotometer (Thermo Scientific, Wilmington, DE, USA).

### 2.4. cDNA Library Construction and RNA Sequencing 

A total of 3 μg RNA per sample was treated with an Epicentre Ribo-zero™ rRNA Removal Kit (Epicentre, Brea, CA, USA) to remove rRNA. The rRNA-free residue was then cleaned up by ethanol precipitation before constructing the RNA-seq libraries. Subsequently, the RNA samples were fragmented and used to synthesize first- and second-strand complementary DNA (cDNA) with random hexamer primers, dNTPs, M-MuLV Reverse Transcriptase (RNaseH-), and DNA Polymerase I. Afterward, the synthetic cDNA fragments were purified using the AMPure XP system (Beckman Coulter, Brea, CA, USA), and the ends were repaired and modified with T4 DNA polymerase and Klenow DNA polymerase to add a single A base and ligate the adapter at the third end of the cDNA fragments. The ligated cDNA products were treated with uracil DNA glycosylase (NEB, Ipswich, MA, USA) to remove the second-strand cDNA. Purified first-strand cDNA was enriched to create the final cDNA library. Lastly, library quality was checked using an Agilent 2100 Bioanalyzer (Agilent, Santa Clara, CA, USA). We sequenced the libraries using Illumina HiSeq 2500 Technology (LC Sciences, Houston, TX, USA).

### 2.5. Sequence Analysis Transcriptome Assembly

Quality control of the RNA-seq reads was performed using FastQC (http://www.bioinformatics.babraham.ac.uk/projects/fastqc/ (accessed on 20 August 2020)). Clean reads were obtained by removing empty reads, adapter sequences, reads with >10% N sequences, and low-quality reads, in which the number of bases with a quality value Q ≤ 10 was >50%. At the same time, the Q30, GC content, and sequence duplication level of the clean data were calculated. Reads that passed the quality control were then mapped to the *Gallus gallus* reference genome (Gallus_gallus-5.0). Based on this, the mapped reads of each sample were assembled with StringTie (v1.3.1) using a reference-based approach [20]. 

### 2.6. Screening and Prediction of DEGs and DE-lncRNAs

Fragments per kilobase of exon per million fragments mapped (FPKM), means the expected number of fragments per kilobase of transcript sequence per million reads sequenced [21]. It takes into account the effects of sequencing depth and gene length on the fragment count and is currently the most commonly used method for estimating gene expression leve l [22]. In this study, transcript abundance was identified by FPKM using Cuffdiff (http://cufflinks.cbcb.umd.edu/manual.html#cuffdiff (accessed on 28 August 2020)) [23]. Here, FPKM was used to calculate the fold change of DEGs between the two groups, and the FPKM of the protein-coding genes in each sample was computed by summing the FPKMs of the transcripts in each gene group. Moreover, we analyzed DEGs by using the edgeR package to calculate the *p*-value that was obtained by multiple hypothesis testing calibrations [24,25]. LncRNAs or protein-coding genes with *p* < 0.05 and log_2_ (fold change) > 1 were assigned as DEGs. 

### 2.7. Construction of the LncRNA-Gene Interaction Network

Previous studies confirm that lncRNAs can regulate gene expression through *cis*-acting and *trans*-acting mechanisms [26]. For each lncRNA locus, the 10 k/100 k upstream and downstream protein-coding genes (without overlap) were first identified as *cis*-target genes. However, the genes that overlapped with the lncRNAs predicted by Lnctar (http://www.cuilab.cn/lnctar (accessed on 12 October 2020)) were selected as *trans*-target genes. To further investigate the interactions between the DE-lncRNAs and their corresponding differentially expressed *cis*- or *trans*-target genes, we constructed an interactive lncRNA-gene network based on their FPKM using Cytoscape 3.8.2 software (http://www.cytoscape.org (accessed on 22 October 2020)). Moreover, we calculated the Pearson correlation coefficient (COR) of each lncRNA and DEG expression value.

### 2.8. GO and Pathway Analysis

GO enrichment analysis of DEGs or lncRNA target genes was implemented using the Molecule Annotation System (MAS) 3.0 (http://bioinfo.capitalbio.com/mas3 (accessed on 2 November 2020)), which is based on the KEGG database (Capital Bio, Beijing). GO terms with *p* < 0.05 were considered significantly enriched by DEGs.

KEGG is a database resource for understanding high-level functions and utilities of a biological system [27], such as the cell, the organism, and the ecosystem, from molecular-level information, especially large-scale molecular datasets generated by genome sequencing and other high-throughput experimental technologies (http://www.genome.jp/kegg/ (accessed on 2 November 2020)). We used KOBAS 3.0 software to test the statistical enrichment of DEGs or lncRNA target genes in KEGG pathways [28].

### 2.9. Analysis of the Expression Levels and Validation by qPCR

For validation via the quantitative real-time polymerase chain reaction (qPCR), single-stranded cDNA was synthesized from 1 µg of total RNA in a final volume of 20 μL according to the manufacturer’s protocol (PrimeScript^TM^ RT reagent Kit with gDNA Eraser, TaKaRa, Dalian, China). The qPCR reactions were performed on an ABI 7500 Fast Real-Time PCR system (Applied Biosystems, Waltham, MA, USA) in a 20 μL volume using Fast Start Universal SYBR Green Master (ROX) (TaKaRa, Dalian, China), and each sample was analyzed in triplicate. The cycling conditions were 95 °C for 30 s, followed by 40 cycles at 95 °C for 5 s and 60 °C for 34 s. A melting curve was obtained at 60–95 °C for each sample amplified. In this study, qPCR primers were designed using the Premier Primer 5.0 software (Premier Biosoft International, San Francisco, CA, USA) and the sequences in GenBank (https://www.ncbi.nlm.nih.gov/ (accessed on 12 February 2021)) and from RNA-seq. The chicken β-actin gene was used as an internal control. The qPCR primer sequences are presented in Table 1.

### 2.10. Statistical Analysis

The results of quantitative expression are presented as the mean ± standard error (SEM), and the significance of the data was tested by a two-tailed paired Student’s *t*-test using SPSS version 20.0 (SPSS Inc., Chicago, IL, USA). The 2^−ΔΔCt^ method was used to analyze the results of qPCR as described [29], and β-actin was used as an internal control to normalize all of the threshold cycle (Ct) values.

## 3. Results

### 3.1. Reads Mapping

In total, we obtained 82,871,160–86,696,604 and 86,622,130–86,688,990 raw reads from the libraries of shell gland tissues of old chickens (*n* = 4) and young chickens (*n* = 4), respectively. Correspondingly, we ultimately obtained 80,510,552–138,847,948 and 84,918,798–85,469,778 clean reads by filtering and removing sequence reads with adapters and low quality, respectively. In addition, the Q30 of each sample was not <90.85%. The quality value of Q30 indicated a 0.1% probability of error base calling during sequencing. Moreover, it was generally accepted that the number of bases with a base quality above Q30 was more than 85%, indicating that the sequencing quality of each sample was high and met the requirements for library construction. Subsequently, we found that >78.77% of the clean reads were completely mapped to the chicken reference genome. The unique mapped reads ranged from 66.24–80.46% of the total mapped reads (Table 2).

### 3.2. Identification and Characterization of LncRNAs

We performed a comparative analysis of the structure of lncRNAs and mRNAs to study the basic features of lncRNAs in the chicken shell gland. This was not just to determine the difference between lncRNAs and mRNAs but also to verify if the predicted lncRNAs were consistent with general characteristics. In this study, the intersection of the Coding Potential Calculator (CPC), Coding-Non-Coding Index (CNCI), and Protein Families Database (PFAM) results yielded 5334 lncRNA transcripts, including the identified conservative lncRNAs (Figure 1A). Interestingly, previous reports indicate that protein-coding transcripts are longer and more conserved than lncRNAs [23]. In agreement with this, we found that the predicated lncRNAs are shorter in length than protein-coding transcripts (Figure 1B) and tend to contain fewer exons (Figure 1C). We also found that the average open reading frame (ORF) length of the predicted lncRNAs was 126 amino acids (aa), which was less than mRNA (687 aa, Figure 1D).

### 3.3. Differential Expression of Predicted LncRNAs and mRNAs in the Eggshell Gland

The expression level of each lncRNA and mRNA was estimated by FPKM using Cuffdiff. To explore similarities and to compare the relationships between the different libraries, we measured the expression patterns of DE-lncRNAs and protein-coding genes by systematic cluster analysis (Figure 2). As a result, we identified 176 lncRNA transcripts that were expressed differentially in the eggshell glands between the old group and young group (Appendix A), and the sequences could be found in the Appendix A. Compared to the young group, 91 lncRNAs were up-regulated, and 85 lncRNAs were down-regulated, in the old group. Among these, the 20 most significantly up-regulated or down-regulated lncRNAs are presented in Table 3 (Figure 2A,B and Table 3). 

Differential expression of mRNAs in shell gland tissues of the old group was also compared to that in the young group. A total of 383 mRNAs were found to be expressed differentially, with 204 up-regulated and 179 down-regulated (Figure 2C,D and Appendix A).

### 3.4. Construction of the LncRNA-mRNA Co-Expression Network

To investigate the questions of whether the functions of DE-lncRNAs are in agreement with their target genes in regulating the chicken eggshell gland, and how do lncRNAs and their target genes interact (*cis* or *trans*), we constructed a co-expression network between DE-lncRNAs and their significantly correlated DE *cis*- and *trans*-target genes using Cytoscape (Figure 3). For the old chicken vs. young chicken comparison, the lncRNA-mRNA co-expression interaction network comprised 37 network nodes and 48 lncRNA-gene connections among 13 DE-lncRNAs and 24 DE-mRNAs. Further analysis revealed that lncRNA upregulated in the shell gland of old chickens increased the expression of *cis*- and *trans*-target genes, except for upregulated *TCONS_03323652* which decreased the *cis*-target gene *LOC422895*. In particular, *TCONS_00181492* and *TCONS_03123639* showed a significant positive correlation with the *cis*-target genes *FGF14* and *GRXCR1*, which are closely associated with calcium and sodium plasma transport [30,31]. In addition, calcium ion transport has a vital role in the eggshell formation, suggesting that *TCONS_00181492* and *TCONS_03123639* play an essential role in regulating eggshell formation.

### 3.5. Enrichment Analysis of the Nearest Neighbor Genes of the lncRNAs

To investigate the functions of the lncRNAs, we predicted their potential *cis* targets. We searched for protein-coding genes 10 kb and 100 kb upstream and downstream of all of the identified lncRNAs. We found 176 lncRNAs that were transcribed close to (<10 kb) 206 neighboring protein-coding genes, and 176 lncRNAs that were transcribed close to (<100 kb) 154 neighboring protein-coding genes (Appendix A). To explore the functions between lncRNAs and their *cis*-regulated target genes, we performed GO analysis. We found 90 GO terms (<10 kb) that were significantly enriched (*p* < 0.05) (Appendix A), and most of these terms were associated with biological processes and molecular functions (Appendix A). In addition, we found 140 GO terms (<100 kb) that were significantly enriched (*p* < 0.05) (Appendix A), and most of these terms were associated with biological processes, molecular functions, and cellular components (Appendix A). For example, the main enriched terms included “protein phosphorylation (GO:0006468)”, “phosphate-containing compound metabolic process (GO:0006796)”, “phosphorus metabolic process (GO:0006793)”, “protein modification process (GO:0006464)”, “ATP binding (GO:0005524)”, “ATP-dependent helicase activity (GO:0008026)”, and “mitochondrial proton-transporting ATP synthase complex (GO:0005753)” (Table 4 and Table 5). Most of them were closely related to the formation of eggshells, which suggests that one of the principal roles of lncRNAs may be to regulate the synthesis and metabolism of organics and minerals. Pathway analysis indicated that *cis*-target genes were significantly enriched in four (<10 kb) and six (<100 kb) KEGG pathways (*p* < 0.05), respectively (Table 6 and Table 7). These data suggest that the function of the shell gland may be regulated by the action of lncRNAs on these neighboring protein-coding genes.

### 3.6. Enrichment Analysis of Co-Expressed Genes of lncRNAs

We also predicted the potential targets of lncRNAs in *trans*-regulatory relationships using co-expression analysis. The COR method was used to analyze the correlation between the lncRNAs and mRNAs in samples, and the main functions of the lncRNAs were predicted using mRNA, with a correlation absolute value >0.95. We found 176 lncRNAs that were transcribed close to 791 protein-coding genes (Appendix A). Functional analysis indicated that the co-expressed genes were significantly enriched in 174 GO terms (95 under biological process, 43 under cellular component, and 36 under molecular function) that encompassed a variety of biological processes (*p* < 0.05) (Appendix A and Appendix A). Importantly, some of the terms were related to organic metabolism and genetic development, including “cellular protein metabolic process (GO:0044267)”, “macromolecule biosynthetic process (GO:0009059)”, “protein metabolic process (GO:0019538)”, “Ras GTPase binding (GO:0017016)”, and “GTPase binding (GO:0051020)” (Table 8). Most of them were associated with organic synthesis and metabolism. The co-expressed genes were significantly enriched in nine KEGG pathways (*p* < 0.05) (Table 9), where the pathways “Salmonella infection” and “AGE-RAGE signaling pathway in diabetic complications” affected the function of the shell gland of aging laying hens. As the disease resistance of aging hens is weakened, the metabolism and synthesis ability of the body is reduced, which leads to a decline in eggshell quality.

### 3.7. Enrichment Analysis of DE-mRNAs

To further understand the biological processes regulated during eggshell formation and to determine which processes are encoded by DEGs, we performed GO and KEGG enrichment analyses on 383 mRNAs. We found 124 GO terms that were significantly enriched (*p* < 0.05) (Appendix A), and most of these terms were associated with biological processes, molecular function, and cellular components (Table 10 and Appendix A). The majority of DEGs were categorized as ion transport in the eggshell gland during the formation of the eggshell. The GO terms included “inorganic anion transport (GO:0015698)”, “inorganic anion transmembrane transporter activity (GO:0015103)”, “anion transmembrane transporter activity (GO:0008509)”, “electron carrier activity (GO:0009055)”, and “calcium ion binding (GO:0005509)”. Thirty-one genes were categorized under these terms, including Glutaredoxin cysteine-rich 1 (*GRXCR1*), and the members (*SLC1A3*, *SLC6A4*, *SLC20A1*, *SLC22A13*, *SLC26A3*, *SLC30A8*, *SLC39A2*, *SLC43A3*, and *SLC45A2*) of the sodium-dependent phosphate transporter family. GO term analysis also revealed some DEGs with possible roles in protein translation and binding. The terms included “protein polymerization (GO:0051258)”, “regulation of G-protein coupled receptor protein signaling pathway (GO:0008277)”, “transcription factor complex (GO:0005667)”, “DNA-directed RNA polymerase II, holoenzyme (GO:0016591)”, and “protein binding, bridging (GO:0030674)”. Several genes were enriched in these terms, most notably *FGF14* and COL5A2. Another important group of DEGs were involved in membrane fiber formation and/or encoded extracellular matrix proteins; “extracellular space (GO:0005615)”, “membrane (GO:0016020)”, and “fibrinogen complex (GO:0005577)” were implicated in this.

In addition, we also found 10 KEGG pathways that were significantly enriched (*p* < 0.05) (Table 11), several of which were related to the function of the shell gland, including “Glycine, serine, and threonine metabolism”, “ABC transporters”, and “Toll-like receptor signaling pathway”. The D-3-phosphoglycerate dehydrogenase (*PHGDH*) gene was significantly enriched in the serine metabolism pathway, and the osteopontin (*SPP1*) gene is a matrix protein that was significantly enriched in the “Toll-like receptor signaling pathway”.

### 3.8. Validation of DE-lncRNAs and -mRNAs

To further validate the reliability and reproducibility of our RNA-seq data, four DE-lncRNAs (*TCONS_00181492*, *TCONS_03234147*, *TCONS_03123639*, and *TCONS_01464392*) and their corresponding target genes (*FGF14*, *COL25A1*, *GRXCR1*, and *GPX8*) related to eggshell quality were randomly selected for qPCR validation. The analysis showed that the expression tendencies of all four lncRNAs and their target genes were extremely concordant with the RNA-seq data, though the absolute fold changes differed between qPCR and RNA-seq (Figure 4 and Appendix A). Appreciably, *TCONS_00181492*, *TCONS_03234147*, and *TCONS_03123639* up-regulated their corresponding target genes, but *TCONS_01464392* down-regulated *GPX8*. These results are consistent with that of the co-expression interaction network, especially for *TCONS_00181492* and *TCONS_03123639* regulating *FGF14* and *GRXCR1*, respectively.

## 4. Discussion

Comparative transcriptome analyses of organs or tissues at different developmental stages can provide valuable insights into the question of how regulatory gene interaction networks control specific biological processes and how diseases can arise [32]. Recently, increasing evidence has confirmed that lncRNAs are important regulatory factors of gene expression, regulating target genes by *cis*-acting (neighboring genes) or *trans*-acting (distant genes) mechanisms [33]. Furthermore, RNA-seq has been performed to provide an extensive lncRNA and gene expression profile in different tissues of livestock and poultry (e.g., cell differentiation and development [34], cancer [35], and skeletal muscle development [36]. Previous studies of the hen uterus transcriptome and gene expression profiling during the formation of the eggshell demonstrate a large number of DEGs that participate in ion transport for eggshell mineralization and the secretion of matrix proteins [37,38,39,40,41]. Most of the previous studies report the roles of mRNAs in the avian eggshell gland, but systematic identification of the functions of lncRNAs remained unclear in the development of the chicken shell gland. Therefore, in this study, we performed transcriptome sequencing of the shell gland of laying hens in the peak and late laying periods and analyzed the DE-lncRNAs and DEGs to reveal their roles in eggshell quality. To the best of our knowledge, this study represents the first systematic genome-wide analysis of lncRNAs and mRNAs in the chicken shell gland, providing a valuable catalog of functional lncRNAs and mRNAs associated with eggshell quality.

In the present study, we developed a highly stringent filtering pipeline to minimize the selection of false positive lncRNAs, which aimed to remove transcripts with evidence of protein-coding potential, and performed co-location mRNA prediction and co-expression mRNA prediction for the lncRNAs obtained from the chicken eggshell gland. Ultimately, we identified 176 DE-lncRNAs and 383 DE-mRNAs. To gain insight into how interactions between DE-lncRNAs and their corresponding target genes regulate shell gland development, we constructed co-expression interaction networks between DE-lncRNAs and their predicted *cis*- and *trans*-target genes. Then, four DE-lncRNAs and their target genes related to eggshell quality were selected for qPCR validation, and the results were consistent with the RNA-seq data, which demonstrated that lncRNA *TCONS_01464392* can target the *GPX8* gene, and they are all down-regulated. LncRNAs *TCONS_00181492*, *TCONS_03234147*, and *TCONS_03123639* target *FGF14*, *COL25A1*, and *GRXCR1*, respectively, and these six genes are up-regulated.

The oviduct of hens is composed of the infundibulum, magnum, isthmus, shell gland, and vagina. Especially, the shell gland is the place where the eggshell is deposited [42]. The formation of the eggshell is a complex process involving the precipitation of calcium carbonate [43]. Mature follicles reach the shell gland and calcify layer by layer. After the mature follicles reach the shell gland, they need to go through the calcification process, and eventually form the eggshell, and the whole process takes about 15–16 h. Approximately 94% of minerals in the eggshell are calcium carbonate, with other inorganic minerals being calcium phosphate, magnesium phosphate, and magnesium carbonate [43]. Previous studies suggest that eggshell calcification requires the interaction of numerous processes, including transcellular and paracellular transport of minerals and the secretion of different matrix proteins [44,45,46]. Particularly, ion transportation plays a crucial role in the process of eggshell formation. The ion channels contribute to the transportation of Ca^2+^ from the plasma to the uterine lumen, which includes Na^+^, Ca^2+^, and K^+^ channels [47]. Moreover, the characteristics of eggshell calcification in poultry are that the body rapidly and massively transports Ca^2+^ from blood to the lumen of the eggshell gland, and a calcium ATPase (calcium pump) is a key enzyme involved in Ca^2+^ transport in the uterus during eggshell formation [48]. Apart from Ca^2+^, inorganic phosphate (Pi) is also essential in the formation of eggshells. Pi is involved in many biological processes, including nucleic acid synthesis, skeletal development, signaling cascades, and tooth mineralization [49,50,51]. More meaningfully, phosphorus participates in the transport mechanism of the calcium pump (calcium ATPase).

In the present study, we conducted GO and KEGG pathway enrichment analyses on DE-mRNAs and DE-lncRNAs and found that the most of identified DEGs were involved in eggshell calcification and cuticularization pathways, such as “inorganic anion transport”, “inorganic anion transmembrane transporter activity”, “phosphate-containing compound metabolic process”, “phosphorus metabolic process”, “protein metabolic process”, “mitochondrial proton-transporting ATP synthase complex”, “proton-transporting ATP synthase complex”, and “calcium ion binding”. Notably, *SPP1* was significantly enriched in the “Toll-like receptor signaling pathway”, and the authors of a previous study suggest that *SPP1* is differentially expressed in the uterus between a low eggshell strength group and a normal eggshell strength group during eggshell formation [41]. In addition, another study indicates that the *PHGDH* gene is highly over-expressed in the isthmus during the deposition of the eggshell membranes [40]. The *PHGDH* gene was also differentially expressed between two groups and enriched in the “Glycine, serine, and threonine metabolism pathway” in this study. Hence, we speculate that the deterioration of eggshell quality during aging in laying hens may be due to disruption of inorganic ion and amino acid transport in the shell gland.

Based on the lncRNA-mRNA co-expression interaction networks, the predicted target gene of lncRNA *TCONS_00181492* is *FGF14*. Prior to this analysis, little was known concerning the association between *FGF14* and lncRNA. *FGF14* is a well-known growth factor belonging to the FGF family. FGF family members possess broad mitogenic and cell survival activities and are involved in a variety of biological processes, including cell growth, embryonic development, tissue repair, morphogenesis, tumor growth, and invasion [52]. Previous work demonstrates that *FGF14* is a functionally relevant component of the neuronal voltage-gated Na^+^ (Nav) channel complex [53], and *FGF14* can also regulate members of the presynaptic Cav2 Ca^2+^ channel family [54]. Simultaneously, there is evidence that the transfer and concentration of Na^+^ can directly affect the transportation of Ca^2+^ and HCO_3_^−^ in the chicken uterus [30]. In the present study, we found that the expression of *FGF14* is up-regulated in the shell gland of chickens in the old group as compared to the young group. The aforementioned studies indicate that the *FGF14* gene plays an important role in chicken growth [31]. The predicted regulatory lncRNA, TCONS_00181492, was significantly more highly expressed in the shell gland in the old group than in the young group and controlled the expression of *FGF14* via *cis*-acting mechanisms. Furthermore, TCONS_00181492 and *FGF14* were positively correlated. Therefore, we had reason to speculate that TCONS_00181492 may regulate shell gland development in the chicken via the *cis*-acting target gene FGF14. Thus, we conjecture that the expression of the target gene FGF14 may be regulated by TCONS_00181492 through *cis*-acting in the shell gland during the aging of laying hens, thereby affecting the ion transport in the shell gland and ultimately leading to the deterioration of eggshell quality.

*COL25A1* was a predicted *cis*-target of TCONS_03234147 that is related to the focal adhesion pathway. Collagen XXV α 1 (*COL25A1*), the extracellular matrix gene, is a collagenous type II transmembrane protein, which was first purified from senile plaques of Alzheimer’s disease (AD) brains [55]. In recent years, work on collagen genes has attracted the attention of many researchers. Previous studies of the hen oviduct transcriptome during eggshell membrane formation identify a large number of differentially expressed collagen genes, such as collagen X (*COL10A1*), collagen I (*COL1A1*), collagen II (*COL2A1*), and collagen III (*COL3A1*) [40]. Moreover, *COL11A1* was also differentially expressed between the normal eggshell strength group and low eggshell strength group in the study integrating transcriptome and genome re-sequencing in the chicken uterus [41]. *TCONS_03234147* and its target gene *COL25A1* were differentially expressed between the two groups in the present study, and their expression was higher in aging hens compared to young hens.

The *GRXCR1* gene is the putative *cis*-target of *TCONS_03123639* in the lncRNAs-genes network. The *GRXCR1* gene encodes an evolutionarily conserved cysteine-rich protein with sequence similarity to the glutaredoxin family of proteins [56]. Recently, research on the function of the *GRXCR1* gene has mostly been focused on diseases [57]. However, the biological function of the *GRXCR1* gene is still rarely reported in livestock and poultry research. Herein, we found that *GRXCR1* was enriched in the ion transport pathway, implying that *GRXCR1* may play an important role in the formation of eggshells. Remarkably, the members (*SLC1A3*, *SLC6A4*, *SLC20A1*, *SLC22A13*, *SLC26A3*, *SLC30A8*, *SLC39A2*, *SLC43A3*, and *SLC45A2*) of the sodium-dependent phosphate transporter (*SLC*) family were also enriched in ion transport pathways (Table 10). Previous studies show that zinc ion transporters include two major families, SLC30 (Solute-Linked Carrier30, also named *ZnT*) and *SLC39* (Solute-Linked Carrier 39, also named *ZIP*). *ZnT* contains 10 transporters of *SLC30A1*-*SLC30A10*, and *ZIP* contains 14 transporters of *SLC39A1*-*SLC39A14*. In our study, the differentially expressed *SLC30A8* and *SLC39A2* genes belong to the ZnT family and *ZIP* family, respectively. Carbonic anhydrase located in the eggshell gland epithelial cells is an important enzyme in the process of eggshell formation, which can reversibly catalyze the hydrolysis of H_2_CO_3_, regulate the concentration of HCO^−^ in the eggshell gland, and then affect the Ca^2+^ transport process and the calcium deposition in the eggshell, changing the quality of the eggshell. Zinc ions are necessary for the activity center of carbonic anhydrase, so zinc can affect the activity of carbonic anhydrase [58]. Moreover, zinc is also a component of alkaline phosphatase, which may regulate some phosphorylated proteins related to the mechanism of eggshell formation and affect the synthesis of calcium carbonate crystals [59]. This provides us a vision for adding appropriate zinc to the diet of aging laying hens, which may reduce the deterioration of eggshell quality.

Through integration analysis of bioinformatics, we found that the differentially expressed *TCONS_01464392* could target the *GPX8* gene, whose expression was extremely significant, and their expression levels were negatively correlated. Glutathione peroxidases (*GPXs*) are enzymes that are present in almost all organisms, with the primary function of limiting peroxide accumulation. In mammals, *GPXs* consist of eight isoforms, but only two members (*GPX7* and *GPX8*) reside in the endoplasmic reticulum [60,61]. A previous study demonstrates that *GPX8* is enriched in mitochondria-associated membranes and can regulate Ca^2+^ storage and fluxes [61]. This indicates that the decline in eggshell quality of aging laying hens may be closely related to down-regulated *GPX8* expression levels.

In conclusion, this research presents the first analysis of lncRNA and mRNA expression in the uterus during the aging of laying hens. A total of 176 DE-lncRNAs and 383 DEGs were identified by comparing the uterus of old and young laying hens. Several novel lncRNAs were revealed. Moreover, functional annotation analysis based on the Gene Ontology (GO) and Kyoto encyclopedia of genes and genomes (KEGG) databases revealed that DE-lncRNAs and DE-mRNAs were significantly enriched in “phosphate-containing compound metabolic process”, “mitochondrial proton-transporting ATP synthase complex”, “inorganic anion transport”, and other terms related to eggshell calcification and cuticularization. These results suggested that lncRNAs in the uterus regulated eggshell quality during aging in laying hens by targeting key genes that modulate ion transport and eggshell calcification. Eight highly associated genes were identified and validated by RT-qPCR, and the results were consistent with the RNA-seq results. These findings provide a solid foundation for future studies on the molecular mechanisms of oviductal senescence in laying hens. The findings laid a solid foundation for future studies on the molecular mechanisms of oviduct aging of laying hens. These findings provide a solid foundation for future studies on the molecular mechanisms of oviduct aging and improve eggshell quality in late-laying hens.

## Figures and Tables

**Figure 1 genes-14-00639-f001:**
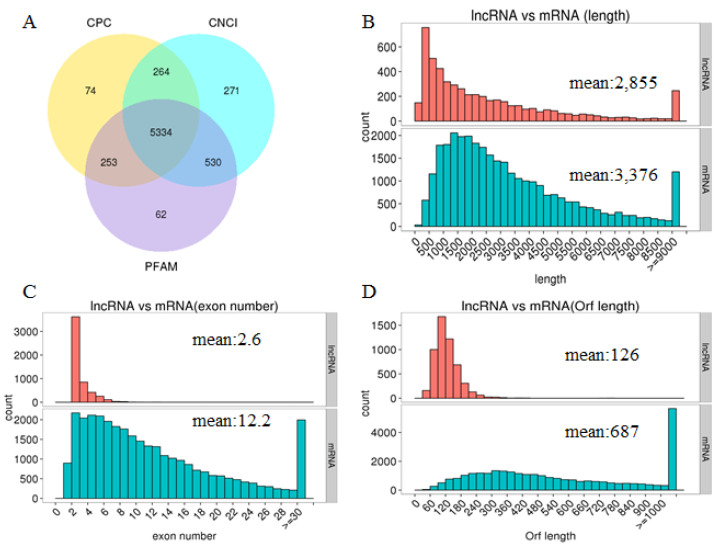
The features of predicted lncRNAs and mRNAs. (**A**) Venn diagram of lncRNAs from the Coding Potential Calculator (CPC), the Coding-Non-Coding Index (CNCI), and Protein Families Database (PFAM). (**B**) Length distribution of lncRNAs and coding transcripts. (**C**) Exon number distribution of lncRNAs and coding transcripts. (**D**) Orf length distribution of lncRNAs and coding transcripts.

**Figure 2 genes-14-00639-f002:**
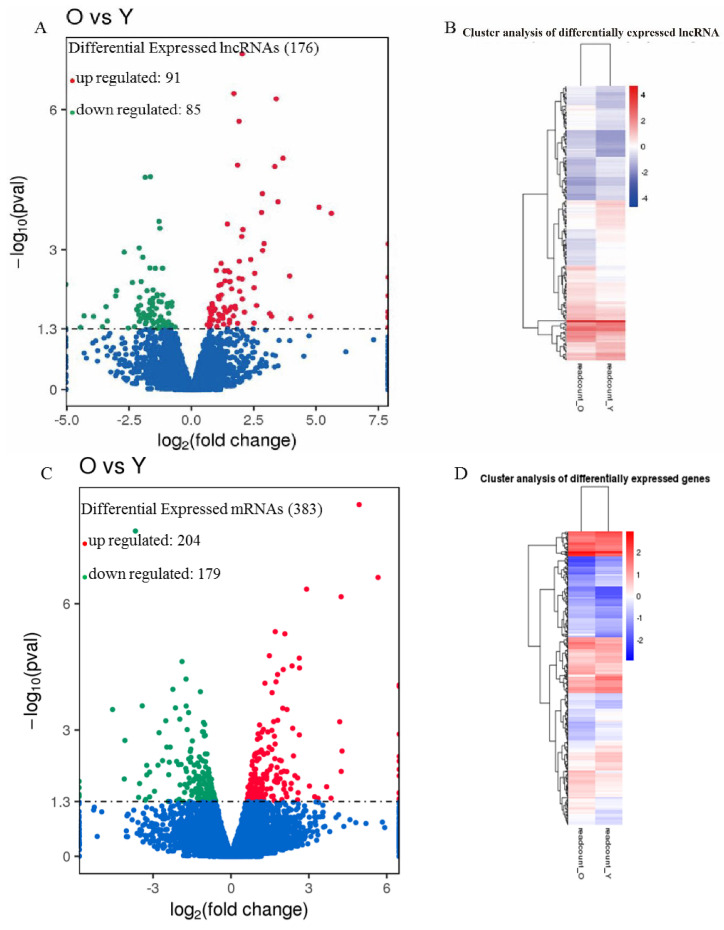
Analyses of DE-lncRNAs and mRNAs in the eggshell gland. (**A**) The volcano plot can intuitively see the overall distribution of the differential transcripts, and the threshold value was set to *p* < 0.05. Blue dots represent that lncRNAs are not significantly differential expressions; Red dots represent relatively high expressions; Green dots represent relatively low expressions. (**B**) A Heatmap of 176 lncRNA expression profiles showed significant expression differences (91 up-regulated and 85 down-regulated). Data were expressed as FPKM, and the red-to-green color gradient indicates from high expression to low expression. (**C**) The volcano plot can intuitively see the overall distribution of the differential genes, and the threshold value was set to *p* < 0.05. Blue dots represent that lncRNAs are not significantly differential expressions; Red dots represent relatively high expressions; Green dots represent relatively low expressions. (**D**) The Heatmap of 383 mRNAs expression profiles showed significant expression differences (204 up-regulated and 179 down-regulated). Data were expressed as FPKM, and the red-to-green color gradient indicates from high expression to low expression.

**Figure 3 genes-14-00639-f003:**
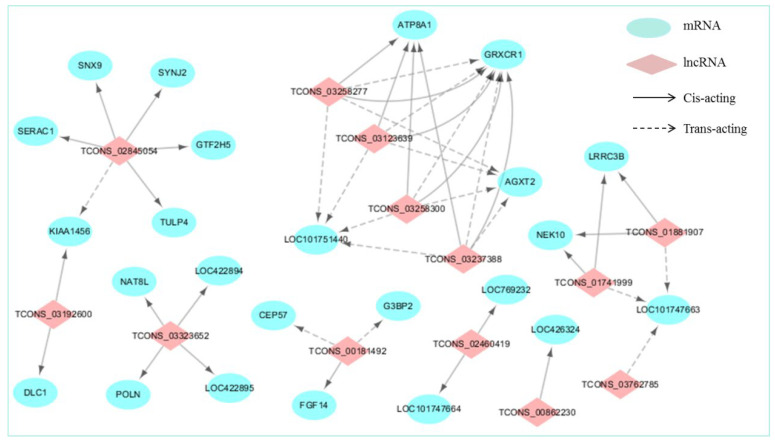
LncRNAs-mRNAs co-expression interaction network. DE-lncRNAs (*p*-adjust < 0.05) and their corresponding differentially expressed *cis*- and *trans*-target genes (*p*-adjust < 0.05) were selected and used to construct a lncRNAs-mRNAs co-expression network. In this network, protein-coding genes are displayed as blue circles, and lncRNA are displayed as pink diamonds. Solid lines mean the interactions between DE-lncRNAs and their corresponding *cis*-target genes, whereas the dashed lines mean interactions between DE-lncRNAs and their corresponding *trans*-target genes.

**Figure 4 genes-14-00639-f004:**
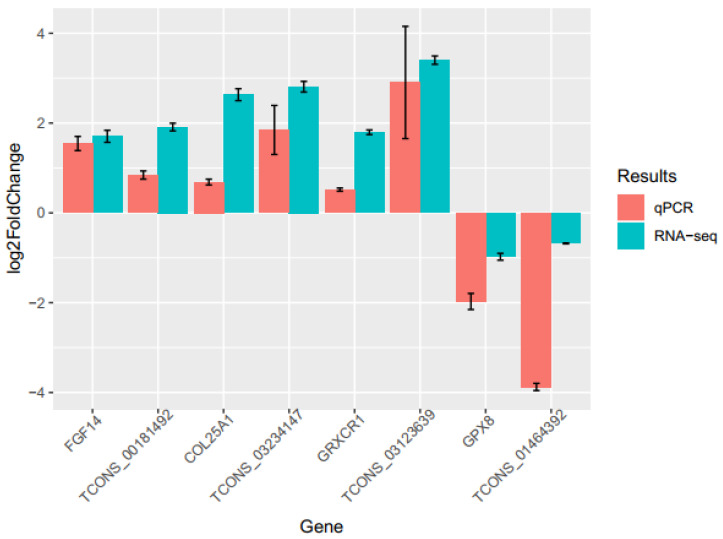
Validation of 4 DE-lncRNAs and their target genes by qPCR. The qPCR results of the DE-lncRNAs and DEGs were compared with their RNA-Seq, respectively. Red: represents qPCR, and Blue: represents RNA-seq.

**Table 1 genes-14-00639-t001:** Primer sequences of qPCR.

Primer Name	Primer Sequences(5′→3′)	Product Size (bp)	Accession ID
*β-actin*	F: CCACCGCAAATGCTTCTAAAC	175	NM_205518.1
	R: AAGACTGCTGCTGACACCTTC		
*FGF14*	F: AATGGCAGTCGTTCAGTAGGATGG	123	NM_204777.1
	R: GCAGAAGGCGGCAGAAGGATC		
*GPX8*	F: CCTCTCACAGCCGCCTATCCTC	111	XM_015277569.2
	R:TCTGAGTTGCAGTAGGCAGAGGAC		
*COL25A1*	F: GACCACCAGGACCACCAGGAC	169	XM_025149941.1
	R: GGCAAGCCAGGTAGTCCAATTCC		
*GRXCR1*	F: TGGTGACTGAGGTACTGCTGGTAG	136	XM_025150281.1
	R: CCTGTAGATGCACGGCTGTTCG		
*TCONS_00181492*	F: GCACTGGACAGCAGCAGCAG	110	_
	R: TAGCCTCACAGCACAGCAGGTAG		
*TCONS_01464392*	F:GTGTCTGTGGCCTCTTACCAATGG	170	_
	R:GCACAGCCAGCATGTAGAAGGTAG		
*TCONS_03234147*	F:TGCCAATAAGCCACCTCAGTCTTC	154	_
	R: GCACCACCTCACTAACCTTCCG		
*TCONS_03123639*	F: GTGTCTGTGGCCTCTTACCAATGG	170	_
	R:GCACAGCCAGCATGTAGAAGGTAG		

Note: F means forward primer; R means reverse primer.

**Table 2 genes-14-00639-t002:** Summary of clean reads mapping to the chicken reference genome.

Sample	Raw Reads	Clean Reads	Q30(%)	Total Mapped Reads	Unique Mapped Reads
O1	82,871,160	80,510,552	90.85	63,421,901 (78.77%)	59,782,215 (74.25%)
O2	86,631,602	85,080,084	91.41	68,637,174 (80.67%)	65,347,267 (76.81%)
O3	86,688,608	85,036,774	92.32	69,540,430 (81.78%)	66,695,503 (78.43%)
O4	86,696,604	85,062,890	92.65	70,184,083 (82.51%)	66,460,473 (78.13%)
Y1	86,688,990	85,370,714	92.72	70,294,495 (82.34%)	56,549,059 (66.24%)
Y2	86,644,390	85,160,146	92.38	69,375,277 (81.46%)	66,196,900 (77.73%)
Y3	86,622,130	85,076,264	92.75	69,868,557 (82.12%)	66,845,243 (78.57%)
Y4	86,674,200	84,918,798	92.57	69,444,146 (81.78%)	66,664,813 (78.5%)

Note: O means old chicken; Y means young chicken.

**Table 3 genes-14-00639-t003:** The top 20 up-regulated or down-regulated lncRNAs.

Transcript ID	Regulation	O/Y(FPKM)	log_2_ Fold Change	*p*-Value
TCONS_01741999	Up	97.03/23.58	2.0409	6.37 × 10^−8^
TCONS_01881907	Up	643.29/197.62	1.7027	4.54 × 10^−7^
TCONS_03123639	Up	130.09/12.31	3.4014	5.89 × 10^−7^
TCONS_00181492	Up	408.1/108.49	1.9114	1.78 × 10^−6^
TCONS_00862230	Up	28.91/2.26	3.6761	1.10 × 10^−5^
TCONS_03323652	Up	203.73/56.3	1.8554	1.55 × 10^−5^
TCONS_03258300	Up	904.76/89.29	3.3411	1.66 × 10^−5^
TCONS_03192600	Down	76/237.39	−1.6431	2.73 × 10^−5^
TCONS_02460419	Down	16.74/60.59	−1.8559	2.82 × 10^−5^
TCONS_02845054	Up	13.56/1.88	2.8483	6.34 × 10^−5^
TCONS_04351227	Up	22.65/2.04	3.472	9.44 × 10^−5^
TCONS_03018542	Up	7.19/0.21	5.12	0.000124
TCONS_03234147	Up	587.16/83.73	2.81	0.00016
TCONS_02608761	Up	11.43/0.23	5.6154	0.000168
TCONS_01162696	Down	46.97/115.45	−1.2975	0.000249
TCONS_03750071	Up	48.36/17.85	1.4382	0.000283
TCONS_01909696	Down	38.98/93.49	−1.2621	0.000349
TCONS_01093310	Up	22.9/5.48	2.0635	0.000371
TCONS_00041803	Up	18.82/4.64	2.0211	0.000525
TCONS_03775965	Up	11.07/1.47	2.9136	0.000745

Note: O and Y respectively indicate the FPKM value of the samples of the old group and young group after standardization; log_2_ foldchange means log_2_ (O/Y).

**Table 4 genes-14-00639-t004:** Gene Ontology (GO) enrichment analysis of differentially expressed protein-coding genes targeted by *cis*-acting (<10 kb) lncRNAs (GO level > 3).

GO Accession	Description	*p*	DEG Item
**Biological process**			
GO:0016310	phosphorylation	0.001224	14
GO:0006468	protein phosphorylation	0.001639	13
GO:0006796	phosphate-containing compound metabolic process	0.002884	15
GO:0006793	phosphorus metabolic process	0.003065	15
GO:0043412	macromolecule modification	0.017898	18
GO:0006464	protein modification process	0.02549	16
**Molecular function**			
GO:0016301	kinase activity	0.001628	17
GO:0000166	nucleotide binding	0.001725	29
GO:0030554	adenyl nucleotide binding	0.001822	25
GO:0005524	ATP binding	0.003026	24
GO:0032559	adenyl ribonucleotide binding	0.003084	24
GO:0008026	ATP-dependent helicase activity	0.003143	5
GO:0070035	purine NTP-dependent helicase activity	0.003143	5
GO:0004672	protein kinase activity	0.004143	13
GO:0017076	purine nucleotide binding	0.004386	26
GO:0003824	catalytic activity	0.004748	68

Note: “DEG item” means the number of DE genes in the category. “GO level > 3” means that each GO term in this table contains more than 3 DE target genes.

**Table 5 genes-14-00639-t005:** Gene Ontology (GO) enrichment analysis of differentially expressed protein-coding genes targeted by *cis*-acting (<100 kb) lncRNAs (GO level > 3).

GO Accession	Description	*p*	DEG Item
**Biological process**			
GO:0006796	phosphate-containing compound metabolic process	5.87 × 10^−5^	44
GO:0006793	phosphorus metabolic process	6.85 × 10^−5^	44
GO:0016310	phosphorylation	0.000143	37
GO:0006468	protein phosphorylation	0.000972	32
GO:0009308	amine metabolic process	0.004619	24
GO:0007275	multicellular organismal development	0.009782	13
GO:0006576	cellular biogenic amine metabolic process	0.009982	6
GO:0009165	nucleotide biosynthetic process	0.010388	13
GO:0006164	purine nucleotide biosynthetic process	0.01094	11
GO:0072522	purine-containing compound biosynthetic process	0.015568	11
**Molecular function**			
GO:0000166	nucleotide binding	0.000177	87
GO:0005488	binding	0.000447	318
GO:0030554	adenyl nucleotide binding	0.000641	71
GO:0017076	purine nucleotide binding	0.000849	78
GO:0005524	ATP binding	0.000976	69
GO:0032559	adenyl ribonucleotide binding	0.001012	69
GO:0032553	ribonucleotide binding	0.0013	76
GO:0032555	purine ribonucleotide binding	0.0013	76
GO:0004867	serine-type endopeptidase inhibitor activity	0.001509	7
GO:0035639	purine ribonucleoside triphosphate binding	0.001815	75
**Cellular component**			
GO:0031012	extracellular matrix	0.000228	19
GO:0005578	proteinaceous extracellular matrix	0.000628	15
GO:0044421	extracellular region part	0.005819	28
GO:0044455	mitochondrial membrane part	0.006614	12
GO:0005753	mitochondrial proton-transporting ATP synthase complex	0.011813	8
GO:0000276	mitochondrial proton-transporting ATP synthase complex, coupling factor F(o)	0.017923	7
GO:0045259	proton-transporting ATP synthase complex	0.022376	8
GO:0045263	proton-transporting ATP synthase complex, coupling factor F(o)	0.031412	7
GO:0005604	basement membrane	0.038506	6
GO:0044420	extracellular matrix part	0.043491	7

**Table 6 genes-14-00639-t006:** KEGG pathway enrichment analysis of differentially expressed protein-coding genes targeted by *cis*-acting (<10 kb) lncRNAs (*p* < 0.05).

KEGG Pathway	Input Number	*p*
Progesterone-mediated oocyte maturation	5	0.007389
Focal adhesion	8	0.008565
Toll-like receptor signaling pathway	4	0.046372
AGE-RAGE signaling pathway in diabetic complications	4	0.049541

Note: “Input number” represents DE-lncRNAs corresponding to the gene number associated with the pathway.

**Table 7 genes-14-00639-t007:** KEGG pathway enrichment analysis of differentially expressed protein-coding genes targeted by *cis*-acting (<100 kb) lncRNAs (*p* < 0.05).

KEGG Pathway	Input Number	*p*
Focal adhesion	17	0.00053242
Drug metabolism-cytochrome P450	5	0.00597761
Metabolism of xenobiotics by cytochrome P450	5	0.00759847
Glutathione metabolism	5	0.01869753
ECM-receptor interaction	7	0.01919077
Progesterone-mediated oocyte maturation	7	0.02296693

**Table 8 genes-14-00639-t008:** Gene Ontology (GO) enrichment analysis of differentially expressed protein-coding genes targeted by *trans*-acting lncRNAs (GO level > 3).

GO Accession	Description	DEG Item	*p*
**Biological process**			
GO:0006412	translation	62	1.94 × 10^−10^
GO:0006996	organelle organization	41	0.000227
GO:0044267	cellular protein metabolic process	110	0.000308
GO:0009059	macromolecule biosynthetic process	157	0.000544
GO:0034645	cellular macromolecule biosynthetic process	156	0.000572
GO:0010467	gene expression	149	0.001188
GO:0019538	protein metabolic process	128	0.001343
GO:2000026	regulation of multicellular organismal development	5	0.003295
GO:0009058	biosynthetic process	182	0.003666
GO:0071841	cellular component organization or biogenesis at the cellular level	56	0.003691
**Molecular function**			
GO:0005201	extracellular matrix structural constituent	6	0.000281
GO:0019899	enzyme binding	11	0.00589
GO:0015035	protein disulfide oxidoreductase activity	5	0.007206
GO:0015036	disulfide oxidoreductase activity	5	0.007206
GO:0017016	Ras GTPase binding	8	0.015394
GO:0031267	small GTPase binding	8	0.015394
GO:0051020	GTPase binding	8	0.015394
GO:0000287	magnesium ion binding	7	0.0216
GO:0008373	sialyltransferase activity	4	0.024643
GO:0016638	oxidoreductase activity, acting on the CH-NH2 group of donors	4	0.026753
**Cellular component**			
GO:0005840	ribosome	50	7.62 × 10^−12^
GO:0005581	collagen	6	3.69 × 10^−5^
GO:0044420	extracellular matrix part	14	5.66 × 10^−5^
GO:0044424	intracellular part	215	8.86 × 10^−5^
GO:0031012	extracellular matrix	24	0.000145
GO:0005605	basal lamina	5	0.001887
GO:0044421	extracellular region part	37	0.002548
GO:0005801	*cis*-Golgi network	4	0.007429
GO:0005606	laminin-1 complex	4	0.007713
GO:0043256	laminin complex	4	0.007713
GO:0005604	basement membrane	8	0.01616
GO:0005874	microtubule	4	0.037275
GO:0033644	host cell membrane	4	0.049272
GO:0044218	other organism cell membrane	4	0.049272
GO:0044279	other organism membrane	4	0.049272

**Table 9 genes-14-00639-t009:** KEGG pathway enrichment analysis of differentially expressed protein-coding genes targeted by *trans*-acting lncRNAs (*p* < 0.05).

KEGG Pathway	Input Number	*p*
Ribosome	61	6.22 × 10^−36^
Focal adhesion	26	6.26 × 10^−6^
ECM-receptor interaction	15	1.35 × 10^−5^
Vascular smooth muscle contraction	11	0.017077
Adipocytokine signaling pathway	8	0.0198
Tight junction	12	0.022353
Salmonella infection	8	0.024636
Adherens junction	8	0.028286
AGE-RAGE signaling pathway in diabetic complications	9	0.038699

**Table 10 genes-14-00639-t010:** Gene Ontology (GO) enrichment analysis of DE-mRNAs (GO level > 3).

GO Terms	*p*	Identified Genes
**Biological process**		
GO:0042221~response to chemical stimulus	0.003154	*SLC43A3*, *NR5A2*, *FDPS*, *GPX8*, *MYO7B*, *CAB39L*, *COL5A2*, *PXDN*, *TLR2-1*
GO:0015698~inorganic anion transport	0.003472	*SLC39A2*, *SLC26A3*, *SLC20A1*, *MYO7B*, *SLC30A8*
GO:0051258~protein polymerization	0.005368	*MYH7*, *KRT6A*, *TUBB6*, *MYO7L3*, *PHYHIPL*, *FGB*
GO:0008277~regulation of G-protein coupled receptor protein signaling pathway	0.00657	*RGS20*, *RGS18*
GO:0009605~response to external stimulus	0.010368	*DMP1*, *SLC43A3*, *CAB39L*, *MYO7B*, *TLR2-1*
**Cellular component**		
GO:0005615~extracellular space	0.002348	*MYH7*, *SOSTDC1*, *KRT6A*, *SLC43A3*, *GNAT3*, *MYO7L3*, *FGB*, *SOGA2*
GO:0005667~transcription factor complex	0.003077	*HIST1H2B8*, *TMEM123*, *HIST1H2B7L3*, *E2F7*, *LAMP3*, *HIST1H2B7L1*
GO:0016591~DNA-directed RNA polymerase II, holoenzyme	0.007195	*HIST1H2B8*, *LAMP3*, *HIST1H2B7L1*, *HIST1H2B7L3*, *TMEM123*
GO:0005577~fibrinogen complex	0.01129	*KRT6A*, *MYH7*, *FGB*, *MYO7L3*
GO:0016020~membrane	0.019963	*SYNPR*, *COL14A1*, *BMPR1B*, *SLC6A4*, *RASL11B*, *ITGB4*, *NMI*, *SOGA2*, *STAT1*, *CCDC59*, *HTR7*, *SLC39A2*, *KCNT2*, *CDHR3*, *SLC1A3*, *SUSD4*, *MUSK*, *MTNR1A*, *PARP14*, *KIAA0319L*, *KRT6A*, *TRPC5*, *ISG122*, *LZTS1*, *ADRB1*, *KRT19*, *GJB1*, *SEC23A*, *IFI27L2*, *TNRC6C*, *SLC43A3*, *FIGF*, *CMPK2*, *SLC45A2*, *ABCC3*, *NPSR1*, *CDH6*, *UGGT2*, *TNR*, *SLC22A13L*, *LAMP3*, *CHRNA7*, *MYO7L3*, *ZCCHC11*, *SLC26A3*, *CPNE1*, *CNR1*, *BCMO1*, *TMEM178B*, *TSPAN13*, *MET*, *DACH2*, *C3AR1*, *CDH23*, *MST1R*, *GPR162*, *C11ORF52*, *SLC20A1*, *CCR8*, *ZFAND3*, *ODZ2*, *YF6*, *TYRP1*, *CCDC110*, *BFSP1*, *HTR1D*, *NDNF*, *KCNC2*, *ANTXR1*, *AFAP1L2*, *KIAA1524*, *SGPP2*, *PDE1C*, *CAMK1D*, *VAMP1*, *TAP2*, *TAP1*, *TLR2-1*,
**Molecular function**		
GO:0015103~inorganic anion transmembrane transporter activity	0.000341	*SLC39A2*, *SLC20A1*, *SLC26A3*, *MYO7B*, *SLC30A8*
GO:0008509~anion transmembrane transporter activity	0.002053	*SLC20A1*, *SLC1A3*, *SLC26A3*, *SLC39A2*, *SLC30A8*, *MYO7B*
GO:0003774~motor activity	0.002122	*MYO7L3*, *KRT6A*, *MYH7*, *KRT19*, *MYO7B*, *LZTS1*, *KIF18A*
GO:0009055~electron carrier activity	0.008214	*LAMP3*, *XDH*, *ZCCHC11*, *GPR162*, *GRXCR1*, *SDHB*
GO:0030674~protein binding, bridging	0.011206	*KRT6A*, *FGF14*, *MYH7*, *FGB*, *MYO7L3*
GO:0008238~exopeptidase activity	0.020203	*CNR1*, *VTN*, *ANTXR1*, *AGBL3*, *COL5A2*, *CPM*
GO:0005509~calcium ion binding	0.021116	*OC3*, *CAPN8*, *CDHR3*, *COMP*, *ERP44*, *CDH6*, *CDHR1*, *MEGF6*, *KIAA0319L*, *MASP2*, *E2F7*, *ANXA5*, *FBLN7*, *CDH23*
GO:0016817~hydrolase activity, acting on acid anhydrides	0.03298	*PLEKHG7*, *KIF18A*, *NLRC5*, *DDX60*, *UGGT2*, *GBP7*, *ABCC3*, *TAP1*, *MYH7*, *TAP2*, *KRT6A*, *RASL11B*, *SMC4*, *MYO7L3*, *FGF14*, *MX1*, *CNR1*, *MYO7B*, *LZTS1*, *KRT19*, *C11ORF52*, *TUBB6*, *GNAT3*, *IFIH1*

**Table 11 genes-14-00639-t011:** KEGG pathway enrichment analysis of DE-mRNAs (*p* < 0.05).

KEGG Pathway	*p*	Identified Genes
Glycine, serine, and threonine metabolism	0.0020501	*AGXT2*, *GLDC*, *TDH*, *AOC3*, *PHGDH*
Phagosome	0.003283	*TUBB6*, *TAP1*, *TLR2-1*, *TAP2*, *CYBB*, *COMP*
ECM-receptor interaction	0.0077845	*LAMB1*, *ITGB4*, *VTN*, *SPP1*, *TNR*, *COMP*
ABC transporters	0.0138254	*TAP2*, *ABCC3*, *TAP1*
Toll-like receptor signaling pathway	0.0156735	*CD86*, *STAT1*, *TLR2-1*, *SPP1*, *IRF7*, *TLR1LA*
Neuroactive ligand-receptor interaction	0.018789	*ADRB1*, *HTR1E*, *LEPR*, *CNR1*, *ADRA2A*, *ADORA1*, *GZMA*, *HTR7*, *C3AR1*, *HTR1D*, *MTNR1A*, *CHRNA7*
Herpes simplex infection	0.021945	*TAP1*, *TLR2-1*, *STAT1*, *TAP2*, *IRF7*, *IFIH1*
Folate biosynthesis	0.0360524	*SPR*, *GCH1*
Glycerolipid metabolism	0.0370195	*LIPG*, *MOGAT1*, *DGAT2*
Tyrosine metabolism	0.0427717	*ALDH3B1*, *AOC3*, *TYRP1*

## Data Availability

The data presented in this study are available upon request from the corresponding author.

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
