# Peer review of "Genome-Wide Analysis of lncRNA and mRNA Expression in the Uterus of Laying Hens during Aging"

_genes, 2023, doi:10.3390/genes14030639_

Round 1

Reviewer 1 Report

The authors present a study where they analyzed and validated the RNA-seq data obtained from shell glands of different aged egg-laying hens. This novel study highlighted the role of lncRNA in regulating eggshell quality. The introduction is well-defined. The methods are also explained in detail. Overall, I am satisfied with the analysis and inference drawn.  However, there are few points where I wish authors could focus and explain a little more:

1. Explain what Q30 is (Table 2) and why is it important. 

2. Total mapped reads was less than 80% in one case. Could authors possible explain the reason?

3. In section 'Construction of lncRNA-mRNA co-expression network', I would urge authors to emphasize on the kind of relationship that lncRNA has with cis/trans targets. For instance, do upregulated lncRNA increase the expression of cis targets and decrease the expression of trans targets or vise versa? Is there any such relationship between lncRNA and target mRNA?

4. In line 268-269, only TCONS_00181492 and TCONS_03123639 lncRNA are mentioned. Is there any specific reason for that? What is the rationale for looking at these lncRNAs specifically? Are these lncRNA studied in literature previously in context to egg shell formation? It would be good if these things are explained.

5. In manuscript, it is mentioned 'one mRNA may correlate with one to four lncRNAs, and one lncRNA may correlate with one to six mRNAs' in lines 270-271.  Again, what is the significance/functional role of such interaction? This need to be explained a bit more.

6. What is meant by co-location genes in line 298?

7. In line 296, ‘Most of them were closely related to the formation of eggshells’. This statement is vague. I think it would help if authors could mention what are these ‘most of the genes’.

 I find the study interesting if these concerns are addressed properly.

Reviewer 2 Report

The article is adequately written. The introduction is sufficiently informative and clear in terms of the stated objectives. It would be appropriate to clarify why the sample size of the laying hens is so small and the connotation this has on the results. The authors consider that the size is sufficient to support the results and conclusions of the study.
